# *Lamp1* Deficiency Enhances Sensitivity to α-Synuclein and Oxidative Stress in *Drosophila* Models of Parkinson Disease

**DOI:** 10.3390/ijms232113078

**Published:** 2022-10-28

**Authors:** Zohra Rahmani, Satya Surabhi, Francisca Rojo-Cortés, Amina Dulac, Andreas Jenny, Serge Birman

**Affiliations:** 1Genes Circuits Rhythms and Neuropathology, Brain Plasticity Unit, UMR 8249, CNRS, ESPCI Paris, PSL University, 75005 Paris, France; 2Department of Developmental and Molecular Biology, Albert Einstein College of Medicine, New York, NY 10461, USA; 3Department of Genetics, Albert Einstein College of Medicine, New York, NY 10461, USA

**Keywords:** Lamp1, α-synuclein, paraquat, Parkinson disease, *Drosophila*

## Abstract

Parkinson disease (PD) is a common neurodegenerative condition affecting people predominantly at old age that is characterized by a progressive loss of midbrain dopaminergic neurons and by the accumulation of α-synuclein-containing intraneuronal inclusions known as Lewy bodies. Defects in cellular degradation processes such as the autophagy-lysosomal pathway are suspected to be involved in PD progression. The mammalian Lysosomal-associated membrane proteins LAMP1 and LAMP2 are transmembrane glycoproteins localized in lysosomes and late endosomes that are involved in autophagosome/lysosome maturation and function. Here, we show that the lack of *Drosophila* Lamp1, the homolog of LAMP1 and LAMP2, severely increased fly susceptibility to paraquat, a pro-oxidant compound known as a potential PD inducer in humans. Moreover, the loss of Lamp1 also exacerbated the progressive locomotor defects induced by the expression of PD-associated mutant α-synuclein A30P (α-synA30P) in dopaminergic neurons. Remarkably, the ubiquitous re-expression of Lamp1 in a mutant context fully suppressed all these defects and conferred significant resistance towards both PD factors above that of wild-type flies. Immunostaining analysis showed that the brain levels of α-synA30P were unexpectedly decreased in young adult *Lamp1*-deficient flies expressing this protein in comparison to non-mutant controls. This suggests that Lamp1 could neutralize α-synuclein toxicity by promoting the formation of non-pathogenic aggregates in neurons. Overall, our findings reveal a novel role for *Drosophila* Lamp1 in protecting against oxidative stress and α-synuclein neurotoxicity in PD models, thus furthering our understanding of the function of its mammalian homologs.

## 1. Introduction

Parkinson disease (PD) is a frequent neurodegenerative disorder characterized by a progressive degeneration of dopaminergic neurons in the midbrain substantia nigra pars compacta and the formation of cytoplasmic inclusions in neurons called Lewy bodies, which are mainly composed of the protein α-synuclein [1]. α-synuclein, encoded by the *SNCA* gene, plays a central role in PD pathogenesis due to its propensity to form neurotoxic oligomers. Oligomerization is enhanced in pathogenic mutant forms such as α-synuclein A30P (α-synA30P), causing familial cases of the disease [2]. PD is clinically characterized by motor symptoms that include resting stiffness and tremors, slow movements and balance problems, resulting in progressive impairment of functional mobility with age. Animal models have been developed to study the neural bases behind these age-related motor impairments. Particularly, *Drosophila melanogaster* is a powerful model organism to better understand the underlying causes of the age-related motor decline observed in PD. The first established *Drosophila* PD model was generated two decades ago by the pan-neuronal expression of human wild-type and mutant forms of α-synuclein [3]. More recently, we have shown that the expression of α-synA30P in a subset of dopaminergic neurons located in the protocerebral anterior medial (PAM) clusters of the fly brain (α-synA30P-PAM model), led to age-related locomotor defects preceded by the degeneration of descending projections from these neurons onto the mushroom bodies (MB) [4].

Lysosomal-associated membrane proteins LAMP1 and LAMP2 are the most abundant proteins of the lysosomal membrane in humans [5]. These transmembrane proteins have two amino-terminal luminal LAMP domains that are heavily N-glycosylated and a short cytosolic C-terminal tail. This glycosylation may act as a protective shield against the lysosomal acid hydrolases [5,6]. Mice deficient for both *Lamp1* and *Lamp2* are embryonic lethal [7] whereas single *Lamp1* or *Lamp2*-deficient mice are viable and fertile [8,9], suggesting that LAMP2 may compensate for the absence of LAMP1 and vice versa and that both proteins share common functions in vivo. However, in contrast to the mild phenotype observed in *Lamp1*-deficient mice, the loss of *Lamp2* causes more severe defects with autophagic vacuole accumulation in several tissues [9] which indicates that, despite their amino acids sequence similarity, LAMP2 has additional unique functions. This is reinforced by the fact that mutations in the *LAMP2* gene cause Danon disease, a rare lysosomal disorder affecting infants, histologically characterized by an accumulation of vacuoles in skeletal and cardiac muscle tissues [10] and an impaired fusion between the lysosomes and the autophagosomes [11,12].

In *Drosophila*, Lamp1 has recently been shown to be the bona fide homolog of the mammalian LAMP1/LAMP2 proteins, although it contains only one LAMP domain upstream of its transmembrane domain [13]. *Drosophila Lamp1* null mutants are fully viable and do not exhibit visible structural or behavioral phenotypes. Larvae display defects in lipid metabolism, with elevated levels of sterols, but no obvious impairments in macroautophagy and endosomal microautophagy (eMI) in the fat body [13]. Here, we address the effects of the *Lamp1* null mutations on two established *Drosophila* models of PD, a sporadic model induced by exposure to the herbicide paraquat and a transgenic model induced by neuronal α-synA30P expression. We show that the lack of *Drosophila* Lamp1 strongly reduced fly survival under paraquat-induced oxidative stress. Moreover, while *Lamp1* deficiency did not by itself induce locomotor impairments, it significantly enhanced age-related locomotor defects in the α-synA30P-PAM model, while apparently decreasing the accumulation of α-synuclein in the brain. The consequences of these findings for the potential molecular function of Lamp1 and its mammalian counterparts are discussed.

## 2. Results

### 2.1. Drosophila Lamp1 Mutants Are Highly Sensitive to Paraquat Induced Oxidative Stress

*Drosophila* exposed to environmental pro-oxidant toxins, such as rotenone or paraquat, reproduce PD features including the selective degeneration of dopaminergic neurons [14,15]. Monitoring the fly survival rate under acute oxidative stress is a practical way to model environmentally-induced PD [15,16,17,18,19]. We have previously shown that the expression of human LAMP2A in neurons protected flies against paraquat-induced oxidative stress by extending their survival [20]. We therefore similarly tested the resistance of *Lamp1* mutants exposed to 20 mM paraquat. Flies homozygous for two independent *Lamp1* null alleles were found to be highly sensitive to the toxin, with an average survival rate of 35.5% ± 12.8% and 42.8% ± 7.0% for *Lamp1*^11B^ and *Lamp1*^6.1^, respectively, vs. 82.7% ± 2.9% for control *w*^1118^ flies after 22 h of paraquat exposure (Figure 1A,B). Importantly, the re-expression of Lamp1 under the control of the ubiquitous *tubulin* driver (*tub-Lamp1*) in the *Lamp1*^6.1^ or *Lamp1*^11B^ mutant backgrounds not only rescued the abnormal susceptibility of *Lamp1*-deficient flies to paraquat but also significantly prolonged their survival rates compared to the wild-type controls (Figure 1A,B). This indicates that the reduced paraquat survival of *Lamp1* mutants was indeed caused by *Lamp1* deficiency and that the ubiquitous expression of Lamp1 can enhance fly resistance to oxidative stress.

### 2.2. Lamp1 Deficiency Accelerates the Age-Related Locomotor Defects Induced by Neuronal α-SynA30P Expression

A commonly used functional test for assessing age-related decline in locomotor performance in *Drosophila* PD models is the startle-induced negative geotaxis (SING) assay (also known as climbing assay) [3,4]. This test allows quantitative scoring of the flies’ climbing activity in a vial or a narrow column in response to a gentle mechanical stimulus. We first checked whether motor function was impaired in *Lamp1* null mutants using the SING test. Remarkably, the age-dependent climbing ability of both *Lamp1*^6.1^ and *Lamp1*^11B^ mutants was found to be comparable to that observed with wild-type control flies (Figure 2A), indicating that the loss of Lamp1 by itself did not induce locomotor defects. As shown previously [4], the expression of α-synA30P under the control of the *NP6510-Gal4* driver (*NP*), which targets a subset of dopaminergic neurons within the PAM cluster, caused a significant progressive reduction in climbing activity compared to the *NP6510-Gal4* and *UAS-SNCA*^A30P^ alone controls (Figure 2B,C). We then found that this age-associated decline was strongly enhanced when α-synA30P was expressed in the absence of Lamp1 (Figure 2B,C for *Lamp1*^6.1^ and *Lamp1*^11B^, respectively). Critically, the ubiquitous re-expression of Lamp1 using *tub-Lamp1* not only rescued the enhanced climbing decline observed in the *Lamp1* mutant flies overexpressing α-synA30P but also fully prevented the age-dependent climbing defects induced by α-synA30P (Figure 2B,C). Taken together, these data demonstrate that while *Lamp1* deficiency does not induce locomotor defects by itself, it strongly increases susceptibility to α-synA30P-mediated progressive locomotor impairments. Furthermore, Lamp1 re-expression is also potently protective against α-synA30P-mediated neurotoxicity.

### 2.3. The Level of Neuronally-Expressed α-SynA30P Is Decreased in Lamp1 Mutant Brains

Because Lamp1 is a lysosomal protein that potentially plays a role in autophagy and we observed that its absence leads to stronger α-synA30P-induced defects, we hypothesized that *Lamp1*-deficient flies might not be able to degrade α-synA30P as well as the controls. To allow a visual assessment of the effect of Lamp1 deficiency on α-synuclein levels in the brain, α-synA30P was expressed under the control of the pan-neuronal driver *nSyb-Gal4* in either wild-type (Figure 3A,B), *Lamp1*^6.1^ (Figure 3C,D) or *Lamp1*^11B^ (Figure 3E,F) mutant flies. In contrast to our expectation, α-synA30P levels detected by immunofluorescence in the brain of 5-day old adults were significantly lower in both *Lamp1* mutants (Figure 3C’’,D’’ for *Lamp1*^6.1^, Figure 3E’’,F’’ for *Lamp1*^11B^) compared to age-matched *w*^1118^ controls (Figure 3A’’,B’’; quantified in Figure 3G). We also immunostained for tyrosine hydroxylase (TH), which marks dopaminergic neurons and observed that its pattern was not affected by the lack of Lamp1 (Figure 3C’,D’ for *Lamp1*^6.1^, Figure 3E’,F’ for *Lamp1*^11B^ respectively vs. Figure 3A’,B’ for controls; quantified in Figure 3H). These results indicate that α-synA30P accumulation is decreased in *Lamp1* mutant brains compared to wild-type flies, which is associated with the higher toxicity of the pathogenic protein.

## 3. Discussion

The LAMP protein family plays an essential role in the autophagy-lysosomal pathway in mammals [5]. *Drosophila* Lamp1 is the homolog of human LAMP1 and LAMP2. However, the lack of both Lamp1 and Lamp2 in mice is embryonically lethal [7], whereas *Drosophila Lamp1* null mutants are fully viable [13]. *Lamp1* mutant adult flies are fertile and display no obvious visible phenotypes, and the Lamp1 protein is dispensable for autophagy in the larval fat body [13]. These differences likely result from the fact that mammalian LAMP1 and LAMP2 have additional functions, such as the role of LAMP2A in chaperone-mediated autophagy (CMA), a process that does not exist in *Drosophila* [13,21]. A better knowledge of the molecular function of *Drosophila* Lamp1 would shed light on changes in lysosomal and LAMP protein function during evolution.

The role of mammalian LAMP proteins in protection against oxidative stress and PD factors has been previously investigated in various models [20,22,23,24]. In the present study, we first sought to examine whether *Drosophila Lamp1* deficiency could affect the susceptibility to oxidative stress induced by the herbicide paraquat. Indeed, we found an enhanced vulnerability to paraquat of two *Lamp1* null mutants compared to *w*^1118^ wild-type flies. Importantly, the re-expression of Lamp1 under the control of a ubiquitous *tubulin* promoter in *Lamp1* mutant background not only abrogated the enhanced susceptibility to paraquat but also promoted a stronger resistance compared to the one observed for *w*^1118^ flies. A possible hypothesis to explain this observation is that Lamp1 expression is regulated and its level limited in some neurons or other cell types. The ubiquitous expression of Lamp1 in *tub-Lamp1*; *Lamp1*^6.1^ or *tub-Lamp1*; *Lamp1*^11B^ flies would not only restore the normal level of Lamp1, but also promote a higher expression of the Lamp1 protein in those cells. This suggests that Lamp1 protects against oxidative stress in flies, which is consistent with the elevated stress resistance of flies expressing human LAMP2A in fly neurons [20].

Although *Lamp1* deficient flies exhibited no age-dependent locomotor defects, we observed that α-synA30P expression in PAM dopaminergic neurons in the *Lamp1* mutant background enhanced the age-related climbing defects observed in this PD model [4], indicating that the lack of *Lamp1* increased the neurotoxicity of α-synA30P. Conversely, we have found that *tub-Lamp1* expression in both *Lamp1*^6.1^ and *Lamp1*^11B^ mutant backgrounds fully suppressed the α-synA30P-induced progressive climbing impairments. These results suggest that the ubiquitous expression of Lamp1 is sufficient to compensate for the age-dependent neurotoxic effects of α-synA30P expressed in PAM dopaminergic neurons. We did not test if re-expressing Lamp1 in the PAM dopaminergic neurons only would be sufficient to rescue the locomotor impairments, or if the rescue resulted from interactions with other cells expressing Lamp1. Nevertheless, our results indicate that Lamp1 expression efficiently protects against α-synA30P in the *Drosophila* PD model.

Unexpectedly, when α-synA30P was expressed pan-neuronally, we immunodetected a lower level of α-synuclein in the brain of both *Lamp1* mutants compared to control flies, although the PD phenotype is much stronger in the absence of Lamp1. This suggests that α-synA30P accumulation in brain cells is higher in the presence of Lamp1 than in its absence. How then could the higher toxicity of α-synA30P in Lamp1-deficient flies be explained? In mammals, α-synuclein is known to be degraded by various intracellular processes that include the autophagy-lysosomal pathway (macroautophagy [25] and CMA [26]) and the ubiquitin-proteasome system (UPS) [25]. When these degradation systems are overwhelmed or deficient, α-synuclein can be neutralized through intracellular storage in large non-pathogenic aggregates such as the Lewy bodies. A potential and so far speculative explanation could be that Lamp1 protects against α-synuclein in flies by promoting the formation of innocuous aggregates of this protein, which would tend to accumulate as they are not easily degraded and which could also be more easily detected by immunostaining than the soluble forms. In the absence of Lamp1, a larger part of α-synA30P would remain monomeric in the cytoplasm or, more likely, owing to its aggregation properties, form soluble oligomers, which are known to be the more toxic species of this protein [27].

In conclusion, our observations indicate that Lamp1 contributes significantly to oxidative stress resistance and the inactivation of neuronal α-synuclein in *Drosophila*. Further study of this protein could therefore open new avenues of research to prevent α-synuclein toxicity and delay PD progression in humans.

## 4. Materials and Methods

### 4.1. Drosophila Lines and Culture Methods

The following fly strains were used (see Appendix A for detailed genotypes): *w*^1118^ (wild-type control), *nSyb-Gal4* from the Bloomington Drosophila stock center (BL-51635), *NP6510-Gal4* [4], and *UAS-SNCA*^A30P^ ([4] and BL-8147). The *Lamp1*^6.1^ and *Lamp1*^11B^ mutants were previously described: *Lamp1*^6.1^ contains frameshift mutations deleting downwards two thirds of the protein, including the transmembrane domain, while in *Lamp1*^11B^, all but the first eight amino acids are deleted [13]. Flies were grown at 25 °C and ~50% humidity with a 12 h light/dark cycle. The food media was a standard cornmeal-yeast agar nutrient medium containing methyl 4-hydroxybenzoate as a mold protector. The *tub-Lamp1* construct was made by first amplifying and cloning the ORF of *Lamp1* from cDNA clone RE72002 with primers lamp-for-Bam (TATGGATCCGCCACCATGTTCGCCAACAAATTGTT) and lamp-rev-Xho (ATACTCGAGTTAGAAGCTCATGTAACCGC) into pSC-A-amp/kan (Agilent Technologies). *Lamp1* then was transferred as BamHI (blunt)/XhoI fragment into the NotI (blunt)/XhoI sites of pCasp-Tub-PA (a gift of Dr. Steve Cohen) [28]. Transgenic injections were performed by Rainbow Transgenic Flies, Newbury Park, CA, USA.

### 4.2. Paraquat Resistance Test

The oxidative stress resistance, scored as the percentage of fly survival after exposure to paraquat, was monitored by dietary ingestion of the herbicide as previously described [16]. For this, non-virgin 7-day old females *Drosophila* (100 females per genotype) were first starved in empty vials for 2 h at 25 °C before being transferred into 35 mm diameter Petri dishes (10 flies per dish) containing two pieces of Whatman paper soaked with 400 µL of 20 mM paraquat (methyl viologen dichloride hydrate; Sigma-Aldrich, St. Louis, MO, USA) diluted in 2% (wt/vol) sucrose (Euromedex). The Petri dishes were then enclosed in a plastic-box with saturating humidity conditions to avoid dehydration and incubated at 25 °C. Fly survival was scored after various indicated times. Starvation and oxidative stress experiments were carried out three times independently and results for each genotype at a given time point correspond to the mean of the scores obtained for the three independent experiments. Statistical analyses were performed with GraphPad Prism using two-way ANOVA with Tukey’s post-hoc pairwise comparisons test. Error bars in figure represent SEM.

### 4.3. Locomotion Assay

The locomotion ability was assayed as previously described using a SING (startle-induced negative geotaxis) test [3,4]. Briefly, 50 adult males of the same genotype were distributed in 5 vertical cone-bottom columns (30 cm length, 1.5 cm diameter, 10 flies per column) and left to rest for 30 min to recover from the transfer process. Each column was then gently tapped down individually in order to startle the flies so that they reacted by climbing upwards. After 1 min, the numbers of flies that reached the top of the column (above 25 cm) as well as the ones that remained at the bottom (below 4 cm) were scored separately. For every single column, three rounds of tests were carried out with a 15 min-interval between each round and results for each group of 5 columns were reported as a performance index (PI) defined as ½[(n_tot_ + n_top_ − n_bot_)/n_tot_], where n_top_ and n_bot_ are the number of flies at the top and the bottom of the column respectively, and n_tot_ is the total number of flies [29]. For each genotype, the complete test was repeated three times independently and the data reported in the graphs correspond to the mean of the scores obtained for the three rounds of test that were carried out at three different periods of time. The SING assay was first performed with 10-day-old flies and repeated each week for a total period of 45 days in order to determine the effect of aging on the fly locomotion ability. Statistical analyses were performed with GraphPad Prism using two-way ANOVA with Dunnett’s correction test for multiple comparisons. Error bars in figures represent SEM.

### 4.4. Adult Brain Immunostaining

Whole-mount brains from female and male adult flies aged 5 days after eclosion were dissected and processed for immunostaining as previously described [16,30]. The following primary antibodies were used: rabbit polyclonal anti-TH (Novus Biologicals, NB300109; 1:1000) and mouse monoclonal antiα-synuclein (Santa Cruz Biotechnology, sc-12767; 1:200). Note that this antibody was raised against a short sequence in the acidic C-terminal region of the protein and is not conformation-specific. Brains were mounted in Prolong Gold Antifade Mountant (Thermo Fisher Scientific, Waltham, MA, USA, P36930) and images were acquired on a Nikon A1R confocal microscope (Nikon Instruments, Tokyo, Japan). Laser, filter and gain settings remained constant within each experiment. Overall fluorescence intensity quantification was determined with the Fiji software by selecting a ROI (region of interest) containing the whole central brain excluding the optic lobes, measuring the integrated density for Z projection images and subtracting background intensity measured from similar sized ROIs lacking tissue. The data were then normalized to respective control values. Six to 15 brains per genotype were analyzed for each independent experiment. Results are represented as mean ± SEM of data obtained in two independent experiments. Statistical analyses were performed with GraphPad Prism using two-way ANOVA with Tukey’s post-hoc pairwise comparisons test. Error bars in figure represent SEM.

## Figures and Tables

**Figure 1 ijms-23-13078-f001:**
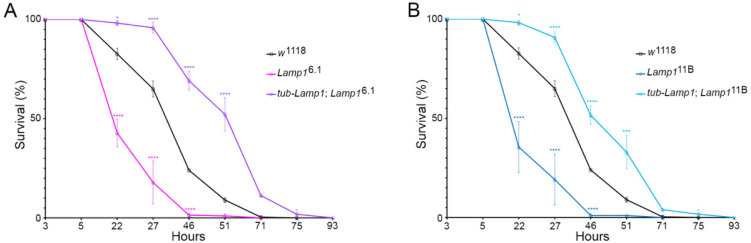
*Drosophila Lamp1* mutants are highly sensitive to oxidative stress. (**A**,**B**) Non-virgin 7-day old *Lamp1*^6.1^ or *Lamp1*^11B^ mutant and *w*^1118^ control female flies were exposed to 20 mM paraquat and the percentages of surviving flies were counted at each indicated time point. Ubiquitous expression of *Lamp1* with a *tub*-*Lamp1* construct rescued the high paraquat sensitivity of the *Lamp1* mutants and conferred further oxidative stress resistance compared to wild-type controls. Detailed genotypes of the strains are reported in Appendix A. Two-way ANOVA with Tukey’s post-hoc test (**** *p* < 0.0001; *** *p* < 0.001; * *p* < 0.05). Error bars represent SEM.

**Figure 2 ijms-23-13078-f002:**
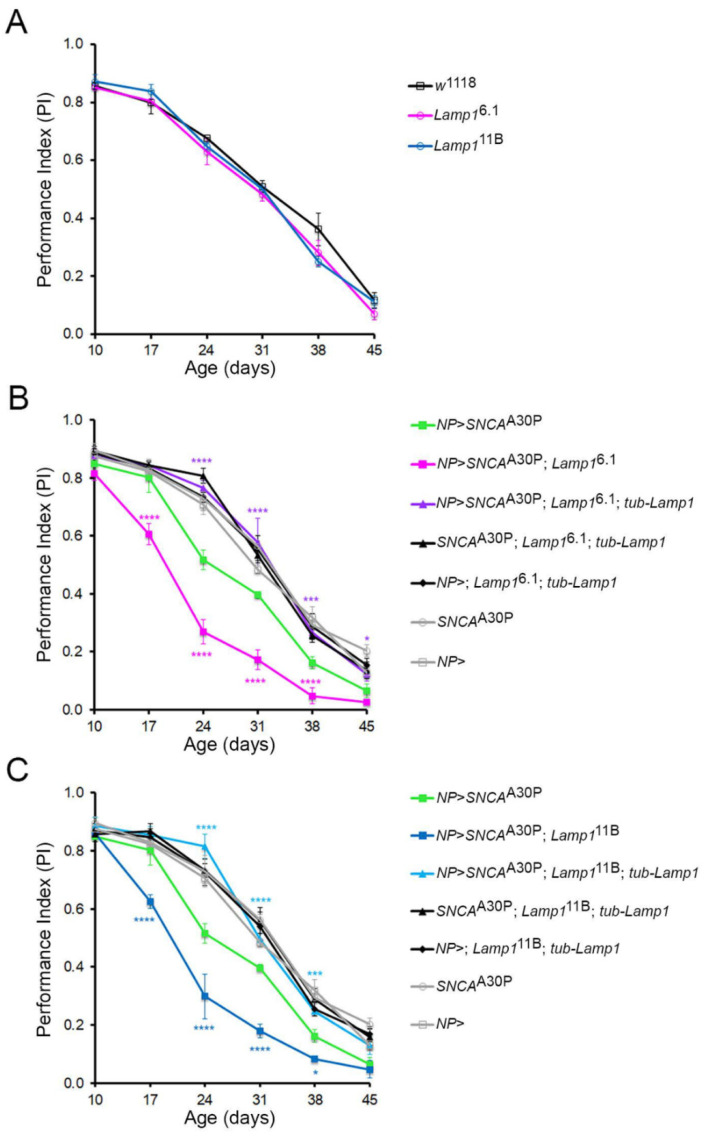
Loss of Lamp1 enhances locomotion defects in the α-synA30P-PAM *Drosophila* PD model. (**A**) The climbing performance of homozygous *Lamp1*^6.1^ and *Lamp1*^11B^ mutant flies is not different from *w*^1118^ control flies throughout life. However, the presence of homozygous *Lamp1*^6.1^ (**B**) and *Lamp1*^11B^ (**C**) mutations significantly worsened the climbing performance of flies overexpressing *SNCA*^A30P^ in PAM dopaminergic neurons using *NP6510-Gal4* (*NP*) over the lifespan of the flies. The co-expression of ubiquitously expressed *tub*-*Lamp1* fully suppressed the deleterious effect of *SNCA*^A30P^ expression in the *Lamp1* mutant backgrounds. Two-way ANOVAs with Dunnett’s post-hoc test for multiple comparisons (**** *p* < 0.0001; *** *p* < 0.001; * *p* < 0.05). Error bars represent SEM.

**Figure 3 ijms-23-13078-f003:**
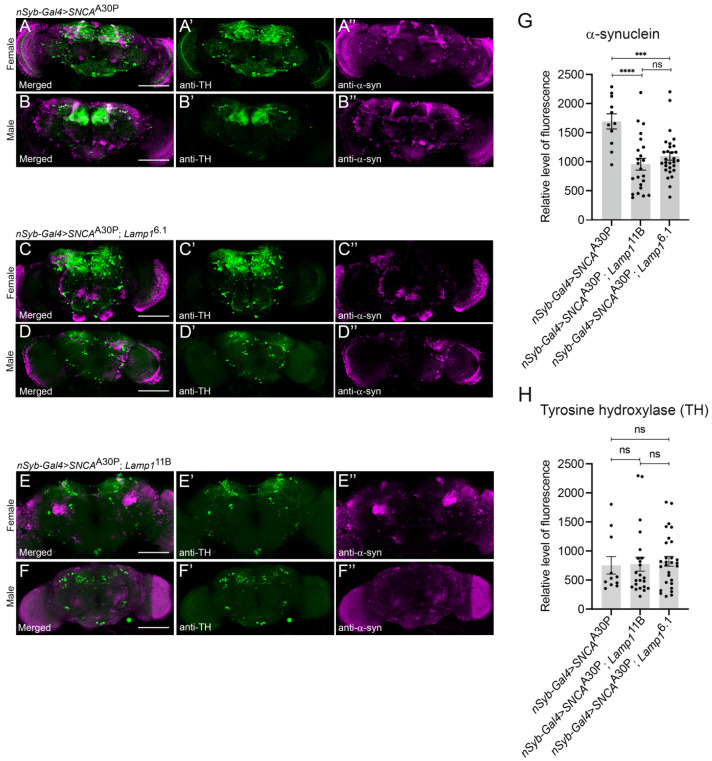
α-Synuclein accumulates less in *Lamp1*-deficient *Drosophila* brains than in controls. (**A**–**F**) Representative confocal images of 5-day old brains from female and male *w*^1118^ control WT flies (**A**,**B**), and *Lamp1*^6.1^ (**C**,**D**) or *Lamp1*^11B^ (**E**,**F**) mutant flies, expressing α-synA30P (*SNCA*^A30P^) pan-neuronally with the driver *nSyb-Gal4*. Maximum projection single channel images show TH (green) (**A’**–**F’**) and α-synuclein (magenta) (**A”**–**F”**), respectively. (**G**,**H**) Quantification of fluorescence intensity shows that the α-synuclein level is significantly lower in both *Lamp1*^6.1^ and *Lamp1*^11B^ mutant brains compared to WT controls (**G**); *nSyb-Gal4 > UAS-SNCA*^A30P^), while TH levels are unaffected at this age (**H**). Bar plots represent the fluorescence intensity levels measured in the whole central brain area excluding the optic lobes. Two-way ANOVA with Tukey’s post-hoc pairwise comparison test (**** *p* < 0.0001; *** *p* < 0.001, ns: not significant). Error bars represent SEM. Scale bars: 100 µm.

## Data Availability

All data presented in this study are available upon request to the corresponding authors.

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
