# Peer review of "Lamp1 Deficiency Enhances Sensitivity to α-Synuclein and Oxidative Stress in Drosophila Models of Parkinson Disease"

_ijms, 2022, doi:10.3390/ijms232113078_

Round 1

Reviewer 1 Report

This is a well written paper with clear results that merit publication. My suggestion to the authors are:

1)   Figure 2 can be improved by breaking up each panel into two. There are far too many genotypes shown in a single panel and it is difficult to distinguish between them. Perhaps controls in one panel and experimental genotypes in another?

2) Figure 3 - it would be greatly helpful if the region of the PAM neurons and MB are also shown at higher magnification.

Author Response

This is a well written paper with clear results that merit publication. My suggestion to the authors are:

1) Figure 2 can be improved by breaking up each panel into two. There are far too many genotypes shown in a single panel and it is difficult to distinguish between them. Perhaps controls in one panel and experimental genotypes in another?

— We understand the reviewer's concern about the difficulty of distinguishing the different genotypes. However, splitting the actual panels into two separate panels for each Lamp1 mutant would make it more difficult for readers to properly appreciate the key message of the data at a glance. Splitting up the data by removing negative controls, or tub-Lamp1 rescue from the baseline phenotype SNCAA30P expression and enhancement by Lamp1 deficiency would make the comparison between those much more difficult. We thus rather provided in the revision larger figure panels for better visibility and would like to emphasize in addition that we also use color coding that reflects the logic of the data.

2) Figure 3 - it would be greatly helpful if the region of the PAM neurons and MB are also shown at higher magnification.

— For the immunostainings in Figure 3, α-synA30P was expressed in all neurons using the driver nSyb-Gal4. The labeling is thus correctly not restricted to the PAM of the MBs in this experiment. Consequently, the quantification was performed on the whole brain (without the optical lobes) as now described in the legend to Figure 3 and page 9 in the Materials and Methods section. This is why we did not include a magnification of the MB area in the figure. The purpose of these scans is also to show that Lamp1 mutants affect α-synA30P accumulation more generally in all neurons of the brain, and not only in the PAM and MB region.

Reviewer 2 Report

Rahmani et al. investigate the role of lysosomal-associated membrane protein LAMP1 in two Drosophila models of Parkinson’s disease: one in which PD is induced by Paraquat, the other with PD-inducing A30P mutation in alpha-synuclein. They report data clearly showing the negative impact of LAMP1 null mutation on the survival rate or progressive course of locomotor impairments in these fly models of PD. They also show that over expression of Lamp1 suppresses or even reverses these negative effects.

It is a concise study of a biological system at basic biology level with high significance in PD. The data seem to be of high quality (I'm not an expert of fly models) and the arguments are sound. The suggested mechanism, that Lamp1 promotes formation of innocuous a-synuclein aggregates, is clearly a speculation but a justified one in my view. I only suggest that they highlight its speculative character. I also suggest that the authors provide a justification for why they use survival rate in model 1 and locomotor activity in model 2 as the readout.

Author Response

Rahmani et al. investigate the role of lysosomal-associated membrane protein LAMP1 in two Drosophila models of Parkinson’s disease: one in which PD is induced by Paraquat, the other with PD-inducing A30P mutation in alpha-synuclein. They report data clearly showing the negative impact of LAMP1 null mutation on the survival rate or progressive course of locomotor impairments in these fly models of PD. They also show that over expression of Lamp1 suppresses or even reverses these negative effects.

It is a concise study of a biological system at basic biology level with high significance in PD. The data seem to be of high quality (I'm not an expert of fly models) and the arguments are sound. The suggested mechanism, that Lamp1 promotes formation of innocuous a-synuclein aggregates, is clearly a speculation but a justified one in my view. I only suggest that they highlight its speculative character.

— We fully agree with the reviewer that the mechanism we proposed to explain the lower levels of a-synuclein immunolabeling in Lamp1 mutants is a suggestion justified by the higher pathogenicity of a-synA30P in these conditions. As requested, we have modified the text to better reflect this on page 7: “A potential, and so far speculative, explanation could be that…”.

I also suggest that the authors provide a justification for why they use survival rate in model 1 and locomotor activity in model 2 as the readout.

— Here we used paraquat to induce acute oxidative stress in flies, which leads quickly to death in wild-type flies, a situation we found to be enhanced by mutation of Lamp1. Monitoring the survival rate has been widely used in such conditions and considered as a convenient way to model environmentally-induced PD in Drosophila (see articles 15 to 20 cited on page 2 in the manuscript). On the other hand, a-synuclein-induced defects are generally monitored using climbing efficiency as the readout since, in contrast to the paraquat model, no externally-induced toxicity would cause death in this case. Startle-induced climbing behavior worsens over lifespan in flies and can reflect, when aggravated, a neurodegenerative phenotype. Therefore, the locomotor activity is used appropriately as a measure of fitness and healthspan in this case. As requested by the reviewer, we have added and clarified sentences on page 2 and 3 to provide justification for the use of these different readouts in the two models.

Reviewer 3 Report

In this study, The authors found the unique physiological significance of Lamp1 in the drosophila model. As the authors state, Lamp1 typically contributes to the expected degradation of proteins. On the other hand, this paper suggests a novel physiological function in regulating α-synuclein degradation and aggregation, thereby suppressing oligomer toxicity. This result is novel and unique, and we believe it is crucial to introduce this phenomenon in the Int J Mol Sci. The research content is also consistent with the purpose of the journal.

Unfortunately, because this submission is a Brief report, the complete picture of the new physiological functions of Lamp1 is not yet clear. In particular, a close examination of the monomeric or multimeric state of α-synuclein and its phosphorylation would be expected. We look forward to furthering the research results.

Minor point

I recommend correcting the character alpha-synuclein "alpha" symbols, garbled into whorls or pteranodons in some places in the manuscript, perhaps due to the different PC environments.

Author Response

In this study, the authors found the unique physiological significance of Lamp1 in the drosophila model. As the authors state, Lamp1 typically contributes to the expected degradation of proteins. On the other hand, this paper suggests a novel physiological function in regulating α-synuclein degradation and aggregation, thereby suppressing oligomer toxicity. This result is novel and unique, and we believe it is crucial to introduce this phenomenon in the Int J Mol Sci. The research content is also consistent with the purpose of the journal.

Unfortunately, because this submission is a Brief report, the complete picture of the new physiological functions of Lamp1 is not yet clear. In particular, a close examination of the monomeric or multimeric state of α-synuclein and its phosphorylation would be expected. We look forward to furthering the research results.

— We do agree with the reviewer that our present work asks for further and more detailed exploration in the future and we will certainly attempt to perform experiments along those lines. However, as stated by the reviewer, we consider it crucial to introduce this phenomenon to a large audience and this is why we chose to publish our work right now as a Brief report.

Minor point

I recommend correcting the character alpha-synuclein "alpha" symbols, garbled into whorls or pteranodons in some places in the manuscript, perhaps due to the different PC environments.

— We apologize to the reviewer for these repeated errors regarding the alpha symbol in the main text. We have corrected this in the revised version of the article.

Reviewer 4 Report

In this communication authors have investigated the importance of Lamp1 in PD flies in combination with A30P alpha synuclein mutant and oxidative stress conditions. The Lamp1 in drosophila is homologous to both lamp 1 and 2 of human counter protein. The Lamp deficiency further correlated with reduced the alpha synuclein accumulation in neurons, however A30P mutation in C elegans model is already reported with less aggregate accumulation as compared to A53T. Can the results observed by authors may be due to the mutation alone? Further whether the antibody used is specific for fibrils/oligomers/monomers of alpha synuclein?

The mutant flies may have an effect on lysosomal membrane integrity, was this aspect looked after by the authors? Any lysosomal membrane marker can be useful for the same.

The alpha synuclein aggregation is initiated when they detach from the membrane bound helical state in to the solution to disordered aggregation prone form. The lamp deficiency may have a direct or indirect effect on initiation of alpha synuclein aggregation, authors comments in this regard will be helpful for the readers.

Author Response

In this communication authors have investigated the importance of Lamp1 in PD flies in combination with A30P alpha synuclein mutant and oxidative stress conditions. The Lamp1 in drosophila is homologous to both lamp 1 and 2 of human counter protein. The Lamp deficiency further correlated with reduced the alpha synuclein accumulation in neurons, however A30P mutation in C elegans model is already reported with less aggregate accumulation as compared to A53T. Can the results observed by authors may be due to the mutation alone?

— We thank the reviewer for bringing to our attention that the A30P mutation of α-synuclein in the C. elegans model was reported to aggregate less than A53T mutation and for raising the concern that our observation may be due to the mutation alone. Here we compared the pathogenicity and accumulation of α-synuclein A30P in wild-type and Lamp1 mutant background. We agree that it would be interesting to test whether Lamp1 deficiency can also reduce accumulation of A53T and other variants of α-synuclein. Although we consider it likely to be the case, we cannot answer this question now, as this would mean to repeat all the work described here for other variants. We believe that this could well be the subject of a further study.

Further whether the antibody used is specific for fibrils/oligomers/monomers of alpha synuclein?

— The antibody we used was not designed to be conformation-specific. We have added a new sentence in the Materials and Methods section on page 9 to explain this: “Note that this antibody was raised against a short sequence in the acidic C-terminal region of the protein and is not conformation-specific, so it should detect the various forms (monomeric, oligomeric or aggregated) of α-synuclein.”

The mutant flies may have an effect on lysosomal membrane integrity, was this aspect looked after by the authors? Any lysosomal membrane marker can be useful for the same.

— Lysosomal integrity would have to be assessed biochemically in purified lysosomal fractions. To the best of our knowledge, lysosomes have never been purified from Drosophila tissue or cells and may not be easily feasible due to limited amounts of starting material. Similarly, we are not aware of another (endo)lysosomal membrane marker for which an antibody exists that could be used for related purposes.

The alpha synuclein aggregation is initiated when they detach from the membrane bound helical state in to the solution to disordered aggregation prone form. The lamp deficiency may have a direct or indirect effect on initiation of alpha synuclein aggregation, authors comments in this regard will be helpful for the readers.

— The comment raised by the reviewer is very interesting. It is quite possible indeed that an interaction between Lamp1 and α-synuclein at the lysosomal surface plays a role in the aggregation process. In the absence of Lamp1, this interaction would be prevented leading to a lower accumulation of neutralized aggregated forms of the toxic protein in neurons. While this would certainly be worth exploring in the future with appropriate tools, we feel that it would be too premature to mention this hypothesis at this stage in our brief report.